# Raspberry Viruses in the Czech Republic, with Identification of a Novel Virus: Raspberry Virus A

**DOI:** 10.3390/v17121597

**Published:** 2025-12-09

**Authors:** Jiunn Luh Tan, Igor Koloniuk, Ondřej Lenz, Jana Veselá, Jaroslava Přibylová, Rostislav Zemek, Josef Špak, Radek Čmejla, Jiří Sedlák, Dag-Ragnar Blystad, Zhibo Hamborg, Jana Fránová

**Affiliations:** 1Institute of Entomology, Biology Centre, Czech Academy of Sciences, 370 05 České Budějovice, Czech Republic; rosta@entu.cas.cz; 2Institute of Plant Molecular Biology, Biology Centre, Czech Academy of Sciences, 370 05 České Budějovice, Czech Republic; koloniuk@umbr.cas.cz (I.K.); lenz@umbr.cas.cz (O.L.); jana.vesela@umbr.cas.cz (J.V.); pribyl@umbr.cas.cz (J.P.); spak@umbr.cas.cz (J.Š.); 3Research and Breeding Institute of Pomology Holovousy Ltd., 508 01 Hořice, Czech Republic; radek.cmejla@vsuo.cz (R.Č.); jiri.sedlak@vsuo.cz (J.S.); 4Division of Biotechnology and Plant Health, Norwegian Institute of Bioeconomy Research, 1433 Ås, Norway; dag-ragnar.blystad@nibio.no (D.-R.B.); zhibo.hamborg@nibio.no (Z.H.)

**Keywords:** *Rubus idaeus*, virus occurrence, raspberry bushy dwarf virus, black raspberry necrosis virus, arthropod vectors, aphids, Sanger sequencing, novel raspberry associated virus A

## Abstract

Although global raspberries production has grown in the past decade, it remains threatened by plant viruses. This study surveyed raspberry viruses and associated arthropods in the Czech Republic between 2021 and 2022 across five regions. A total of 257 plant and 151 arthropod samples were tested using RT-(q)PCR for 12 viruses listed in the EPPO Certification scheme, plus raspberry leaf blotch virus (RLBV) and a novel virus, tentatively named raspberry-associated virus A (RaVA). Raspberry bushy dwarf virus (RBDV) was most prevalent (51.8%), followed by black raspberry necrosis virus (BRNV, 42.0%) and raspberry leaf mottle virus (RLMV, 28.4%). Four viruses—arabis mosaic virus, apple mosaic virus, strawberry latent ringspot virus, raspberry ringspot virus—were not detected. RBDV was also identified in *Sambucus nigra*, a new host, while mixed RLBV and RaVA infection was found in wild *Rubus occidentalis*. RLBV was experimentally transmitted to *Nicotiana occidentalis* 37B in the presence of *Phyllocoptes gracilis*. Seven of 39 arthropod species carried viruses, but only two—*Amphorophora rubi idaei* and *Aphis idaei*—are known vectors. PCR amplicons from 92 isolates were sequenced, revealing high variability in several viruses. These findings offer new insights but highlight the need for continued monitoring and research.

## 1. Introduction

The global production of raspberries has been increasing over the past decade, reaching its highest levels in recent years (2020–2023), with yields exceeding 900 thousand tons [1]. This growth is driven by raspberries’ high economic value, owing to their versatility and nutritional benefits [2]. However, raspberry (*Rubus idaeus* L.) is known to be affected by at least 30 fungal pathogens, 5 bacterial species, strains of two 16Sr phytoplasma groups (16SrV-E, 16Sr XII-A) and 24 plant viruses [3,4,5,6,7,8,9]. In addition, raspberry is also infested by many invertebrate herbivores, some of which are recognized or suspected vectors of raspberry viruses [3]. For instance, the large raspberry aphid, *Amphorophora rubi idaei* (Börner) (Hemiptera: Aphididae), is known to transmit at least three raspberry viruses: black raspberry necrosis virus (BRNV, *Sadwavirus rubi*, family *Secoviridae*), raspberry leaf mottle virus (RLMV, *Closterovirus macularubi,* family *Closteroviridae*), and Rubus yellow net virus (RYNV, *Badnavirus reterubi*, family *Caulimoviridae*) [3,10,11]. Among all plant pathogens, viruses pose a major threat to raspberries production, given the fact that their infection remains incurable to date, impacting both the yield and longevity of the plant [12].

In the Czech Republic, research on raspberry viruses and their vectors has a considerable knowledge gap. Early investigations relied on symptomatology [13,14,15], and later on less sensitive serological methods, such as the double diffusion test in agar and DAS-ELISA [16,17,18]. Only five viruses—cherry leaf roll virus (CLRV, *Nepovirus avii*, family *Secoviridae*), arabis mosaic virus (ArMV, *Nepovirus arabis*, family *Secoviridae*), cucumber mosaic virus (CMV, *Cucumovirus CMV*, family *Bromoviridae*), strawberry latent ringspot virus (SLRSV, *Stralarivirus fragariae*, family *Secoviridae*), and tomato black ring virus (TBRV, *Nepovirus nigranuli*, family *Secoviridae*)—were tested using these serological methods. This is because these techniques depended on the availability of antibodies for each target virus, thus limiting the scope of virus screening in *Rubus*. Furthermore, the early studies focused only on wild-growing populations [16,17,18]. A follow-up study expanded the scope of virus screening by including an additional virus—raspberry ringspot virus (RpRSV, *Nepovirus rubi*, family *Secoviridae*)—and examined both cultivated and wild raspberry populations. However, most detected viruses could not be confirmed using the verification methods available at that time, such as mechanical inoculation and transmission electron microscopy [19], raising questions about detection accuracy due to methodological limitations. Raspberry bushy dwarf virus (RBDV, *Idaeovirus rubi*, family *Mayoviridae*) has also been reported in the Czech Republic since 1994, with subsequent studies addressing its biology and epidemiology [20,21,22,23]. Since then, no surveillance of raspberry viruses was conducted until recently, despite advances in molecular diagnostic methods such as polymerase chain reaction (PCR) and Sanger sequencing.

The most recent study was conducted by Valentová et al. [24], using more advanced detection method of quantitative polymerase chain reaction (qPCR). However, that study was limited to seven viruses and did not include arthropod vectors. It also covered only a narrow geographical region—selected sites in eastern and northern Bohemia. Our recent work using high-throughput sequencing (HTS), an advanced method that enables simultaneous screening of known and novel viruses, led to the discovery of two new viruses—raspberry enamovirus 1 (RaEV1) [6] and raspberry rubodvirus 1 (RaRV1) [9]—further emphasizing the importance of comprehensive raspberry viruses surveillance in the Czech Republic. Therefore, our current study aims to fill these gaps by screening a broader range of viruses, covering all viruses listed in the EPPO Certification scheme for *Rubus* [25] and the not-yet-listed raspberry leaf blotch virus (RLBV, *Emaravirus idaeobati*, family *Fimoviridae*). In addition, the study has also expanded the geographic scope by including sampling sites across different regions of the Czech Republic and multiple sources of raspberry (i.e., commercial plantations, wild populations, private gardens, and garden retailers), and has incorporated virus detection in potential arthropod vectors. In summary, we aimed to address several research gaps:The lack of comprehensive raspberry virus surveillance in the Czech Republic, including all raspberry viruses listed in the EPPO Certification scheme for *Rubus* and RLBV—as well as the need to expand both the sources of plant material and the geographic regions surveyed.Limited screening of potential arthropod vectors, which is crucial for understanding virus–vector dynamics, especially in the context of rapid environmental change.The ambiguities surrounding earlier detections of viruses using serological methods, which can now be clarified with more sensitive and precise diagnostic tools such as RT-(q)PCR and Sanger sequencing.Lack of knowledge about the genetic diversity of viruses affecting raspberry in the Czech Republic.

To date, no study of this scale has been conducted on raspberry in the Czech Republic, as no surveillance studies was carried out or published between the earlier research and the recent works by Valentová et al. [24], Koloniuk et al. [6], and Lenz et al. [9]. It is also possibly the first study on global scale to conduct such extensive simultaneous screening of both raspberry viruses and their putative vectors, as similar studies typically focused on one or a few selected viruses, and rarely included arthropods simultaneously (for example [26,27,28,29,30,31]). Therefore, this study expands knowledge of raspberry virus diversity in the Czech Republic, while providing insight into potential virus–vector relationships in this crop. This is essential for developing effective management strategies under current rapidly changing environmental conditions, enabling timely interventions to prevent major yield losses. Although this study is conducted in the Czech Republic, its findings will be valuable for updating the EPPO Certification scheme and other virus disease management policies, benefiting raspberry-producing countries worldwide.

## 2. Materials and Methods

### 2.1. The Sampling

Raspberry plants and arthropods were collected from May to September in 2021 and from May to August in 2022 across 24 sites in 5 regions of the Czech Republic, namely Hradec Králové (HK), Liberec (L), Pardubice (PR), Central Bohemia (CB), and South Bohemia (SB) (Table 1, Figure 1). The locations are listed in Appendix A. Sample sites represented: (1) commercial plantations, (2) garden retailers, (3) private gardens, and (4) plants growing in the wild, such as forest and woodland patches. The largest commercial raspberry plantations, Břežany and Dobré Pole (CB) were approximately six years old, and cultivation was under non-woven fabrics tunnels. The open field plantation in Vyhnánov, Kbelnice and Synkov (HK) were 20, 10, and 2 years old, respectively. The most common cultivars were Polka, Sugana, Enrosadira and Canby. Most of the private garden raspberries were of unknown cultivars and of undetermined age. Wild raspberries were found far away from the cultivated fields; therefore, there were no inclusions of cultivated cultivars among the wild-growing raspberry plants.

Raspberry plants showing suspected viral symptoms or plants infested with potential virus vectors, regardless of symptoms, were preferentially sampled. In commercial plantations (Břežany and Dobré Pole, CB; Synkov and Kbelnice, HK), asymptomatic and arthropod-free plants were also sampled for control purposes. In smaller plantations (Vyhnánov, HK), no asymptomatic plants were found and thus no control samples were available. Plant samples, along with a small piece of water-saturated tissue paper, were placed into polyethylene zip-lock bags, which were then kept in a cool box during transportation to the laboratory. Representative examples of the sampled leaves and shoots are shown in Figure 2. A total of 257 raspberry plant samples were collected (Appendix A).

In addition, a wild-growing elderberry shrub (*Sambucus nigra* L.) with symptoms of light green rings on leaves (Appendix A) and growing about 10 m away of raspberry plants in a private garden (Nový Bydžov, HK), was also sampled. *Rubus occidentalis* wild-growing plant (Volanice, HK) showing severe leaf blotch symptoms was included in virus screening too.

### 2.2. Arthropod Samples

Arthropods were collected using a purposive sampling method, targeting arthropod groups with a potential to act as virus vectors based on their diet and biology. The sampling was carried out concurrently with plant collection at the same 24 sites (Table 1). The arthropods were collected either directly from plant surfaces using a mouth aspirator—typically for mobile arthropods (e.g., *Orius* spp.)—or together with leaves taken from both asymptomatic plants and those showing symptoms of virus infection—typically for less mobile arthropods (e.g., aphids and spider mites). The samples were placed into polyethylene bags with moist tissue paper to prevent desiccation during transportation to laboratory. The collected arthropods, except thrips and mites, were identified to the family level, and to the genus or species level when possible, using a binocular stereo microscope (Technival 2, Carl Zeiss, Jena, Germany), without being killed, to preserve the samples for subsequent RNA isolation. Thrips and mites were identified only using molecular techniques due to their minute size, which required them to be killed for morphological identification. A total of 151 arthropod samples were collected (Appendix A).

### 2.3. Detection of Raspberry Viruses Using Molecular Methods

A total of 14 raspberry viruses were tested in the plant samples (Table 2), including a novel virus species, tentatively named raspberry-associated virus A (RaVA). All the tested viruses were listed in the European and Mediterranean Plant Protection Organization (EPPO) Certification scheme for *Rubus* [25], except for RLBV and RaVA. Their detection in 257 raspberry plants, *R. occidentalis* and *S. nigra* was carried out either by reverse transcriptase polymerase chain reaction (RT-PCR) or reverse transcriptase quantitative polymerase chain reaction (RT-qPCR) using previously reported or newly designed primers (Appendix A). The six economically most important viruses infecting raspberry: BRNV, RBDV, RLBV, RLMV, raspberry vein chlorosis virus (RVCV, *Alphacytorhabdovirus alpharubi*, family *Rhabdoviridae*), and RYNV were also tested in the collected arthropods. For a detailed overview of the viruses found in all samples, as well as the primers, and PCR conditions, refer to Appendix A.

#### 2.3.1. RNA Isolation and cDNA Synthesis

Total RNA was isolated from approximately 100 mg of fresh, chilled (4 °C), or frozen (−20 °C) raspberry leaves, from at least one to a group of smaller living arthropods, or from the head and thorax of bigger arthropods using a Ribospin™ Plant Kit (GeneAll Biotechnology, Seoul, Republic of Korea), following the manufacturer’s protocol. The leaf areas exhibiting the suspected virus symptoms were preferred. The quantity of RNA was measured by NanoDrop 1000b spectrophotometer (NanoDrop Technologies, Wilmington, DE, USA). The cDNA was synthesized using M-MLV Reverse Transcriptase (Invitrogen, Waltham, MA, USA). In each 20 μL reaction, about 0.5 μg of RNA was used for plants or 0.01–0.2 μg of RNA was used for arthropods as a template. The synthesized cDNA was used in RT-PCR or RT-qPCR. To verify the quality of the prepared cDNA from the tested plants and to exclude the presence of viruses in the diet of insects, samples were tested using Atropa Nad2.1a/2.2.b primers [32].

#### 2.3.2. Virus Detection by RT-PCR

In RT-PCR, one microliter of the cDNA was mixed with 10 μL of 2× PPP Master Mix (Top-Bio, Vestec, Czech Republic) or SapphireAmp^®^ Fast PCR Master Mix (Takara Bio Inc., Kusatsu, Japan) in the case of RaVA, 7 μL of PCR-grade H_2_O, and the corresponding primers at 0.5 μM (1 μL each) (Appendix A). Positive controls for BRNV, RBDV, RLBV, RLMV, RVCV, and RYNV were obtained as cDNA from the Norwegian Institute of Bioeconomy Research (NIBIO), Norway. An RNA preparation from the lamina tissue of the raspberry cv. Polka (isolate A503, not included in this study), where RaVA was previously identified via HTS, was employed as positive control for this virus. The no-template control was created with the same reaction mixtures excluding the cDNA templates.

The PCR product (4 μL) was subjected to electrophoresis on a 1% agarose gel pre-stained with GelRed DNA stain (Biotium, Hayward, CA, USA), and DNA bands were visualized using a UV transilluminator.

#### 2.3.3. Virus Detection by Two-Step RT-qPCR

All RT-qPCR assays were performed using a CFX96 real-time PCR detection system (Bio-Rad, Hercules, CA, USA). Each 10 µL RT-qPCR reaction mixture contained 5 µL of tenfold-diluted cDNA (prepared as previously described in Section 2.3.1, corresponding to approximately 12.5 ng of RNA from the original input), 0.25 µL of primers mixture (10 mM stock, yielding a final concentration of 250 nM for both forward and reverse primer, see Appendix A), 2.75 μL of nuclease-free water, and 2 µL of 5′ HOT FIREPol EvaGreen RT-qPCR Mix Plus (Solis BioDyne, Tartu, Estonia). For certain targets, to avoid excessive primer degeneration, several primers targeting the same region for diverse isolates were synthesized as an alternative to using degenerate primers (for example SLRSV; Appendix A). By organizing sequences into identity-based clusters and designing cluster-specific primers, we eliminated the need for degenerate primers as the presence of multiple nucleotide permutations at degenerate positions means that only a small fraction of the primer molecules will perfectly match the target sequence. That may lead to reduced binding affinity and PCR efficiency. These cluster-specific primers were designated with a slash notation (e.g., 3844/3845:3842/3843—3844/3845 forward mixture and 3842/3843 reverse mixture for CLRV detection) and combined to yield a final directional mixture with final reaction concentration of 250 nM for either forward or reverse mixture.

The reactions were executed with a three-step cycling protocol: an initial denaturation at 95 °C for 12 min, followed by 40 cycles of 95 °C for 10 s, 58 °C for 20 s, and 72 °C for 20 s. A dissociation curve analysis was conducted by gradually increasing the temperature from 65 °C to 95 °C in 0.5 °C increments for 5 s each, to confirm primer amplification specificity and to check for the presence of potential primer dimers, indicated by a single peak. No template controls and positive controls were incorporated to assess potential cross-contamination and the presence of genomic DNA. Data analysis was performed using Bio-Rad CFX Maestro 1.1 (Bio-Rad).

The positive controls for ApMV, ArMV, CLRV, CMV, SLRSV and TBRV were RNA isolated from lyophilized plant tissue previously known to be infected with the respective virus deposited in the virus collection at Biology Centre CAS, the Czech Republic. Positive control for RpRSV was obtained as cDNA from NIBIO.

#### 2.3.4. Sanger Sequencing

Selected RT-PCR and qPCR products were bidirectionally Sanger-sequenced using the respective PCR primers (Eurofins Genomics, Luxembourg), following purification with Expin Combo GP mini (GeneAll Biotechnology, Seoul, Republic of Korea). To confirm the arthropod identification, 103 selected arthropod samples: cDNAs amplified through PCR using primers specific to the cytochrome C oxidase subunit I (COI) [33] were Sanger sequenced. In the case of *Phyllocoptes gracilis* (Nalepa) (Acari: Eriophyidae), cDNA prepared from raspberry leaves (A929, B284) occupied with this mite was used as a template, as the RNA extraction includes material from minute arthropods such as eriophyid mites. Sequence analyses were performed with Geneious Prime software, version 2025.1.2 (Biomatters, Auckland, New Zealand). Primer– and quality–trimmed sequences were used for downstream analyses. All sequences were identified using the Basic Local Alignment Search Tool (BLAST, version 2.15.0) provided by the National Center for Biotechnology Information (NCBI) (accessed on 1 May 2025).

### 2.4. Transmission of Raspberry Leaf Blotch Virus to Nicotiana occidentalis 37B

The first attempt to transmit RLBV to *N. occidentalis* 37B was conducted in June 2021. Three leaf pieces (size: ~0.5 × 1.5 cm) exhibiting severe leaf blotch symptoms were taken from a RLBV-positive raspberry plant (A929), which was colonized by *Aphis idaei* van der Goot (Hemiptera: Aphididae) and the raspberry leaf and bud mite *P. gracilis*, and placed on the leaves of one *N. occidentalis* 37B plant. The second attempt took place in early August 2025. The inoculum source was another RLBV-positive raspberry plant (B284), colonized by only *P. gracilis*. The presence of *P. gracilis* on raspberry leaves was examined using a binocular stereo microscope (Technival 2, Carl Zeiss, Jena, Germany), and its identity was confirmed by molecular barcoding of the COI gene. Three pieces of symptomatic leaves were placed on the leaves of *N. occidentalis* (*n* = 15) and *Physalis floridana* (*n* = 15) plants. In both attempts, the plants were kept in the cages inside an insect-proof greenhouse and cultivated at 25 °C under a 16L:8D photoperiod (L—light, D—darkness). Symptoms development was monitored daily for at least 120 days following transfer of symptomatic leaf pieces to recipient plants. Total RNA isolated from a batch of six aphids (two individuals from each leaf fragment) and from the symptomatic recipient *N. occidentalis* plants was tested for the presence of all 14 viruses, with RLBV tested multiple times during the second attempt.

## 3. Results

### 3.1. Symptoms Associated with Viruses on the Collected Samples

The presence of viruses was detected in visually healthy plants, in plants with very mild and easily overlooked symptoms, and in plants with pronounced symptoms. Conversely, none of the tested viruses were detected in two plants with severe symptoms—sample B62, exhibiting dark green veins and yellowing, and sample A963, showing mosaic and dwarfing. Of the 70 shrubs that showed no symptoms, no virus was detected in 36 samples. However, latent infection with one to three viruses was detected in the remaining 34 shrubs. RBDV and BRNV (each found in 6 plants) and RLMV (found in 5 plants) were the most common viruses present without symptoms. Interestingly, no symptoms were observed at the time of sampling in four plants in which co-infection with three viruses was detected (CLRV, RBDV, and RYNV in one plant; and RaVA, RBDV, and RYNV in three plants).

Regarding the presence of arthropods, of the 49 asymptomatic samples in which arthropods were found, 29 were positive for at least one virus (59.2%), while no virus was detected in 20 plants (40.8%). Among the 21 asymptomatic samples where no arthropods were found, 5 were positive for one or two viruses (23.8%), and no virus was detected in 16 samples (76.2%). Colonies of *A. idaei* were found on 27 asymptomatic plants, 16 of which were virus positive. Individual *A. rubi idaei* were found scattered on 12 asymptomatic plants, 5 of which tested positive for either BRNV or RLMV, and one for both viruses simultaneously.

At the sites in Břežany and Dobré Pole, it was difficult to find plants showing clear signs of viral infection. Plants with symptoms were most often located at the edges of fields. In Břežany, the most common symptom observed was severe yellowing with dark green veins, occasionally plants with veinal chlorotic mottle were found. Of the 17 asymptomatic shrubs tested, no virus was detected in 3 plants. In the remaining plants, RBDV, RYNV, and RaVA were detected either alone or in combination. In Dobré Pole, the most common symptom was also yellowing, followed by dot mosaic, and in rare cases, dark green mosaic. Of the 22 asymptomatic plants, no virus was identified in 11 shrubs, while in the remaining plants, RLMV prevailed, either alone or in combination with RBDV, CLRV, and BRNV.

In Vyhnánov, all plants were found with symptoms, including stunting, leaf yellowing or reddening, abnormal flower shape, and, in rare cases, mosaic and vein clearing. The most frequently detected viruses were RBDV, BRNV, and RLMV, with CLRV and RaVA also occurring relatively often.

Severe leaf blotch symptoms were observed in the Synkov commercial plantation, where RLBV was confirmed. No virus was detected in four asymptomatic plants. Nearly all plants at the Kbelnice commercial plantation exhibited leaf blotch symptoms of varying intensity, with vein clearing observed in rare cases. In the plants showing leaf-blotch symptoms, RLBV was detected either alone or in combination with RBDV or RaVA. RLBV was also detected alone in one plant without visible symptoms. RVCV and RBDV were identified in a plant exhibiting vein-clearing symptoms.

Distinct types of symptoms were also observed in wild vegetation at the respective locations. In the HK region, leaf blotch, vein clearing, and necrosis predominated, with RLBV, RVCV, and TBRV detected in symptomatic plants. In the SB region, wild plants with severe mosaic were found, mostly infected with BRNV. Yellowing was associated with the presence of BRNV, TBRV, and RBDV. Leaf blotch symptoms corresponded with the detection of RLBV. Plants exhibiting a dashes pattern around veins and chlorotic mottle were also found infected with BRNV. BRNV and RBDV were detected in one of the two plants with veinal chlorotic mottle. Multiple infection of BRNV, RBDV, and RaVA was detected in the other. Among 19 symptomless wild plants tested, 15 were virus-free, and BRNV was found in the remaining four plants.

The symptoms observed in all plants and the viruses detected in them are listed in Appendix A. Representative examples of these symptoms are shown in Appendix A.

### 3.2. The Prevalence of Viruses in Raspberry Plants

Out of the 14 screened viruses, ApMV, ArMV, RpRSV, and SLRSV were not detected in any of the 257 samples. The prevalence of the remaining 10 viruses is summarized in Table 3, with the most prevalent being RBDV (51.8%), BRNV (42.0%), and RLMV (28.4%). Eight of the detected viruses were found in three locality types (commercial fields, private gardens, and wild), while BRNV and RYNV were also detected in plants from a garden retailer (although only two plants from this source were analyzed in total). In contrast, RLMV was detected only in commercial plantations.

The prevalence of viruses varied between sampling sites, as demonstrated by the comparison of three commercial plantations (Figure 3), where 48–61 samples were screened per a site.

At the Vyhnánov plantation, no virus-free plants were found—all 61 tested plants carried at least one virus. Only 2% of these plants showed a single infection, while the majority harbored three or more viruses (Figure 4 and Figure 5). This contrasted sharply with the other two plantations (Břežany and Dobré Pole) and wild plant samples, where a substantial proportion of plants (13–37%) tested negative, and single infections ranged from 29% to 43% (Figure 4 and Figure 5). Nevertheless, shrubs at the latter two plantations were only six years old, while those at Vyhnánov were approximately 20 years old.

BRNV emerged as the most prevalent virus in wild samples, occurring frequently in both single and mixed infections (Figure 5). In contrast, samples from other sources displayed a diverse array of viral combinations, with no single virus predominating.

Differences in virus prevalence were also observed between cultivars grown on the same plantation (Appendix A). At Břežany, RBDV and RYNV were more prevalent in the Polka cultivar (93.8% and 50.0%, respectively) compared to Sugana (68.8% and 12.5%, respectively)—for differences in Dobré Pole see Appendix A. Since this study was not designed for direct cultivar comparisons, and the cultivars tested varied across the analyzed sites, cultivar-specific susceptibility to particular viruses cannot be reliably concluded. Nonetheless, the observed differences in RBDV prevalence between ‘Sugana’ shrubs of the same age (six years) at Břežany and Dobré Pole (68.8% vs. 14.3%, respectively), suggest that factors such as sampling site, or the origin of the mother plant stock may have a greater influence than the cultivar itself.

Additionally, RBDV was also detected in *S. nigra* using all three used primers pairs (Appendix A).

### 3.3. Prevalence of the Novel RaVA Virus

The novel RaVA was notably abundant in the commercial plantation in Břežany (39.6%; 19/48), where it was detected only in cv. Polka (59.4%; 19/32), but not in cv. Sugana (0.0%; 0/16). Ten positive plants were found in Vyhnánov (16.4%), and no positive plant was recorded in Dobré Pole. Raspberry bushes from gardens also showed a high prevalence of infection (61.1%; 11/18). Specifically, in the neighboring Hradec Králové and Pardubice regions, positive plants were found in all five gardens surveyed (90.0%; 9/10). The RaVA was least abundant in wild plants (7.6%; 5/66), although all three wild-growing plants from the České Budějovice 2 site were positive. In addition to red raspberry, RaVA was detected also in a wild black raspberry (*R. occidentalis*) plant in co-infection with RLBV.

### 3.4. The Presence of Viruses in Arthropods

A total of 39 arthropod species were identified through molecular techniques and/or morphological methods. Eleven species were found to carry at least one of the six tested raspberry viruses (Table 4). Among them, the large raspberry aphid, *A. rubi idaei* (Figure 6A,B), the small raspberry aphid, *A. idaei* (Figure 6C–E), and the lucerne bug, *Adelphocoris lineolatus* (Goeze) (Hemiptera: Miridae), were found to harbor at least three of these viruses: BRNV, RBDV and RLMV. Among them, BRNV and RBDV were the most frequently detected viruses in both arthropods and raspberry plants. Among the virus-carrying arthropods, four species were predators or omnivores.

The other 20 species of phytophagous arthropods and eight non-herbivorous arthropods collected on raspberry were negative for all six tested viruses (Appendix A).

Details of the individual virus detection in arthropods are provided in Appendix A.

### 3.5. Sequence Variability of Detected Viruses

In total, PCR amplicons from 77 isolates of 10 virus species from raspberry bushes, 15 virus isolates from arthropod tissues (Appendix A), one RBDV isolate from *S. nigra* and RLBV isolates from *R. occidentalis* and *N. occidentalis* 37B were Sanger sequenced. The highest sequence variability among the Czech virus isolates was recorded for RLMV (76.6–100%; 8 isolates evaluated), BRNV (80.0–99.1%; 7 isolates evaluated) and RYNV (81.6–100%; 20 isolates evaluated) in the compared regions. The least variable were TBRV (100%; 2 isolates evaluated), RLBV (99.1–100%; 12 isolates evaluated), RaVA (98.4–100%; 7 isolates evaluated), and RBDV (96.0–100%; 10 isolates evaluated). Thirty-six selected sequences were deposited in GenBank (Acc. No. PX549241–PX549276; out of them PX549250 and PX549251 are sequences of RBDV detected in *S. nigra*).

For the details, see Appendix B.

### 3.6. Raspberry Leaf Blotch Virus Experimental Transmission to Nicotiana occidentalis 37B

In the first attempt to transmit RLBV to *N. occidentalis* 37B (conducted in 2021), mild chlorosis and leaf deformation were observed in the recipient plant 14 days after arthropod establishment, followed by mosaic symptoms, further deformation, and necrosis (Figure 7A–C). In the second attempt (conducted in 2025), mild mosaic symptoms were observed on 1 of the 15 *N. occidentalis* plants 20 days after the start of the experiment. Seven days later, in the same plant, necrosis and leaf deformation developed on newly formed leaves. Two months post-inoculation, mild chlorosis and necrotic spots were also observed on the new leaves, followed by dark green mosaic symptoms on other leaves approximately three months post-inoculation (Figure 7D–F). However, no symptoms were observed on *P. floridana* plants.

The presence of RLBV in *N. occidentalis* 37B was detected in symptomatic leaves by RT-PCR and confirmed by Sanger sequencing 27 days post-inoculation in the first attempt. The obtained sequence (530 nt) was identical to that of the RLBV isolate from the donor raspberry bush (Acc. No. PX549271). Infected *N. occidentalis* 37B leaf materials were lyophilized and deposited in UPOC collection funded by the Ministry of Agriculture of the Czech Republic as a part of The National Programme on Conservation and Utilization of Microbial Genetic Resources and Invertebrates of Agricultural Importance (https://www.microbes.cz/indexangl.html; accessed on 20 October 2025) under an accession number UPOC-VIR-049. No RLBV or other viruses were detected in aphids feeding on the infected raspberry leaves used for transmission. In the second attempt, RT-PCR results were negative for RLBV at 25 days post-inoculation (dpi) when the leaf showing mild mosaic symptoms was tested (Figure 5D), and positive at 70 dpi using RNA isolated from a newly growing leaf displaying dark green mosaic symptoms. At 106 dpi, RT-PCR tested positive using extracts from both chlorotic and asymptomatic leaves. However, another asymptomatic leaf on a neighboring shoot tested negative at 106 dpi. None of the other 13 viruses screened were detected in the raspberry donor plants, or *N. occidentalis* 37B plants. Given that *P. gracilis* (Acc. No. PX444998) was present in both RLBV transmission experiments and since successful transmission occurred in both cases, it is very likely to be the vector of RLBV. To our knowledge, *N. occidentalis* 37B represents a new experimental host of RLBV.

## 4. Discussion

### 4.1. Symptoms and Associated Viruses

The study demonstrates the diversity of symptoms and the presence of viruses in the tested plants. Although symptoms alone cannot definitively identify which viruses are present, certain symptom types can often be associated with specific viruses. For example, this is observed in RLBV, which manifested in most plants as severe yellow blotching, as previously described [34,35]. However, RLBV was also detected alone in asymptomatic raspberry bush, or in co-infections with RBDV, BRNV, and RYNV in plant exhibiting severe yellowing and dark green veins. When RVCV was present, vein clearing may be observed on the leaves. In the presence of RBDV, yellow sections on the leaves with distinctly dark green veins were frequently observed, particularly in co-infections with other viruses. BRNV was often found in asymptomatic plants but, particularly in wild plants, was associated with pronounced mosaic, veinal chlorotic mottle, or chlorotic mottle. Similarly, RLMV rarely caused visible symptoms, although systemic dot mosaic was occasionally observed. In long-term cultivated plants infected with three or more viruses, dwarfism was the dominant symptom. The symptoms observed in this study are largely consistent with previous descriptions of these viruses [36]. However, to our knowledge, the occurrence of a light green pattern around veins has not previously been reported in raspberry. In addition, the molecular detection methods used enabled the detection of viruses even in asymptomatic plants, especially those infested with aphids or other arthropods.

### 4.2. Prevalence of Previously Known Viruses in Raspberry

The importance of this study lies in its thorough assessment of all the 12 viruses recommended by the EPPO Certification scheme for *Rubus* [25], except for raspberry yellow spot virus and raspberry leaf spot virus, both of which are likely isolates of RLMV [37,38]. The employed techniques (RT-PCR or RT-qPCR) are also more sensitive compared to ELISA used in the certification scheme [25]. The newly designed primers developed in this study contribute to improved diagnostics of the 9 out of 14 tested viruses.

One of the most important raspberry diseases is the raspberry mosaic disease (RMD) complex, which is caused by three viruses: BRNV, RLMV, and RYNV. These viruses, together with RBDV, were also among the most frequently detected in our study. They are vectored by economically important aphid that feed exclusively on raspberry and blackberry, *A. rubi idaei* (in Europe) [3]. BRNV is considered one of the most widespread raspberry viruses, largely because it is difficult to detect, as its infection is typically symptomless or causes only mild symptoms in this crop. However, it kills shoot tips, followed by chlorotic mottling or mosaic in black raspberry [39,40]. Recently, BRNV associated with severe symptoms has been reported in the raspberry cultivar ‘Glen Ample’ in Norway [41], and distinct mosaic and dwarf were noted in a wild-growing raspberry bush at Vrábče in this study (e.g., isolate A702; only BRNV detected by HTS). The previously described veinal chlorotic mottle [41] was also observed in wild-growing plants infected with BRNV alone or in co-infection with RBDV. These symptoms may result from several factors, including plant physiological traits, host susceptibility to viruses, duration of infection, variability among virus strains, or co-infection with other viruses. BRNV was also detected in a purchased, asymptomatic raspberry plant of cultivar ‘Summer Chef’. Although Valentová et al. [24] did not detect BRNV in commercial plantations, they nevertheless found it in a private collection (17% of infected plants), private gardens (8%), purchased plants (4%), and in a source of propagation material (11%). This finding supports previous reports that BRNV infections can circulate undetected in symptomless plants, which could serve as the virus reservoir.

Although RYNV was not the most prevalent in our study, it was reported as the most prevalent in the study by Valentová et al. [24]. They attributed the high prevalence of RYNV to the lack of an appropriate detection method prior to 2002, which may have allowed the virus to circulate widely in the community—resulting the higher prevalence rates in plants sold at garden retailers and later found in private gardens. Furthermore, the considerable genetic diversity of RYNV isolates, and the recent discovery of endogenous RYNV associated with a large number of commercial red raspberry cultivars, remain a significant challenge for detection [42,43]. It is also likely that real-time PCR using the Rotor-Gene Q (Qiagen) instrument with newly designed primers and probes [44] provides a more sensitive detection system than the RT-PCR method used in the present study.

The high prevalence of RBDV may be attributed to its nature as a pollen-borne virus. It was absent from purchased plants originating from garden retailers, likely due to the propagation materials are regularly tested in accordance with the Czech Decree No. 96/2018 Coll.—Decree on Breeding Plantations and Propagation Material of Fruit Genera and Species and Its Placing on the Market that adopts EU legislation, and any RBDV-positive plants are immediately removed. RBDV was the most prominent in commercial plantations and was detected in seven of the eleven virus-positive arthropod species collected during the study. Among these arthropods, except for *A. rubi idaei* and *A. idaei*, all are known to feed on pollen, suggesting efficient transmission through pollen. This is particularly concerning, as these include generalist predators—such as *Orius* spp. and *Typhlodromus* (*Typhlodromus*) *pyri* Scheuten (Acari: Phytoseiidae)—whose diets include non-prey food sources such as pollen and nectar, which may contribute to pollen-mediated virus transmission. In addition, the increasing use of bee pollination in raspberry production could facilitate the spread of pollen-borne viruses, such as RBDV [45,46]. While the transmission of pollen-borne viruses has been associated with pollinators such as bees and phytophagous insects, the role of generalist predators is often overlooked and warrants further investigation [47]. Furthermore, RBDV was also detected in elderberry (*Sambucus nigra*), indicating its potential to spread to a new host species. Although the exact cause of observed symptoms (i.e., light green rings) remains uncertain, the sample was co-infected with elderberry virus A and elderberry virus B (identified via HTS), both of which are known to infect elderberry [48].

In addition, two other important raspberry viruses—RVCV and RLBV—often yield inconsistent results. Although RVCV is widely known to be vectored by *A. idaei* [49], transmission experiments have shown inconsistent results, suggesting that retention time may likely depend on the duration of feeding by *A. idaei* after virus acquisition. The virus was observed to be lost more quickly when feeding time increased [50]. RVCV was previously spread through infected planting stock [51], but its current low prevalence in commercial fields (3.5%) and absence in purchased plants from garden retailers suggest that efforts to screen for virus-free planting stock have been effective. However, further studies on the vector–RVCV interaction are recommended. Earlier studies indicated that densely packed *A. idaei* colonies were more efficient at transmitting RVCV [50], but these findings date back more than six decades. With advancement of molecular technologies in recent years, this interaction should be reinvestigated in greater detail to anticipate potential changes in virus–vector dynamics. A particularly notable finding in this study was that two different primer pairs amplified what appear to be distinct RVCV variants from the same plant. This highlights the need for caution in primer selection for RVCV detection, as the choice of primer may directly influence both detection results and the interpretation of viral diversity.

RLBV is a relatively recently discovered virus, and currently, *P. gracilis* is the only recognized vector [34,35]. The successful transmission of RLBV to *N. occidentalis* 37B in this study, in the presence of *P. gracilis*, strongly supports its role as the vector. However, one of the greatest challenges in studying RLBV still lies in its inconsistent occurrence and the variability of infection symptoms. RLBV-positive samples do not always exhibit the characteristic leaf blotch symptoms typically associated with the virus [35,52], which was also confirmed in our study. In some cases, infected plants remain symptomless, making detection and diagnosis difficult. Interestingly, we observed inconsistency in RLBV distribution even in the herbaceous host *N. occidentalis* 37B. Specifically, symptomatic leaves tested negative and positive by RT-PCR at 25 dpi and 70 dpi, respectively, whereas at 106 dpi, both symptomatic and asymptomatic leaves from one shoot were positive, and asymptomatic leaves from another shoot were negative. Although the results of this study do not indicate that RLBV is widespread in the Czech Republic, a high prevalence has been reported in Serbia [29]. In addition, characteristic leaf blotch symptoms consistent with RLBV infection were observed farm-wide in two commercial raspberry fields in the HK region (i.e., Kbelnice and Synkov) in 2023, where some affected plants eventually declined. Importantly, this study has demonstrated the potential of RLBV to infect plants other than raspberry, suggesting that it may pose a broader threat than initially assumed. The existence of multiple RLBV isolates, which may require different primers for accurate detection, cannot be ruled out, as this is common among plant viruses [30,35]. These findings highlight the need for further research on RLBV and the fundamental challenges associated with its study.

In addition to the six major raspberry viruses, the previously uncertain presence of CLRV in *Rubus* in the Czech Republic about three decades ago [19], has now been confirmed. Previously reported symptoms of infected *Rubus* plants include stunted, distorted leaves with severe chlorotic mottling, as well as ring and line patterns, which can lead to significant economic losses [4,53]. Given its pollen-borne nature and confirmed presence in wild populations, greater awareness of CLRV in raspberries is needed. The role of flower-visiting arthropods, including beneficial insects, in the transmission of pollen-borne CLRV must be further researched. CMV and TBRV were also detected, each with an abundance of below 5%, and are considered less concerning. In the Czech Republic, the presence of TBRV on raspberry was previously reported by Špak et al. [19], with similarly low prevalence in DAS-ELISA. This suggests a limited spread in raspberry over the past 30 years, likely due to strict regulatory controls on virus-free propagation material (Decree No. 96/2018 Coll.—Decree on Breeding Plantations and Propagation Material of Fruit Genera and Species and Its Placing on the Market that adopts EU legislation). CMV, transmitted by more than 80 species of aphids and via seed [54], is rarely and inefficiently spread between raspberry plants [55,56]. Although the relevant transmission efficiency studies were conducted more than three decades ago, its low prevalence in our study supports those earlier findings. Furthermore, the highly efficient CMV vectors, *Myzus persicae* (Sulzer) (Hemiptera: Aphididae) and *Aphis gossypii* Glover (Hemiptera: Aphididae), were absent in this study. Notably, a mixed infection of CMV and RYNV was detected in an asymptomatic purchased plant of the cultivar ‘Polka’ of unknown origin. However, CMV was present in less than 1% of all samples, highlighting its limited spread.

The remaining four viruses—ApMV, ArMV, RpRSV, and SLRSV—were not detected in any of the tested samples. Although ApMV has a wide host range [57], its presence in raspberry within the EPPO region has been rare, whereas it has been reported as widespread in raspberry and blackberry in North America [58]. The occurrence of the other three viruses—ArMV, RpRSV and SLRSV—was documented in the Czech Republic approximately three decades ago, with only RpRSV showing higher overall prevalence. However, the DAS-ELISA results could not be validated at the time, as mechanical inoculation of the virus was unsuccessful [19]. While ArMV was not tested, both RpRSV and SLRSV were also absent in raspberry samples from a recent study by Valentová et al. [24]. Since neither current monitoring nor past study [19] have found any widespread plant dieback or crop losses typically associated with nematode-borne viruses [36], the occurrence of these viruses in raspberry within our territory is likely rare.

In addition to the prevalence of individual viruses, it is important to note that co-infections are pervasive in both commercial plantations and wild populations. This is particularly evident on farms with long-term continuous cultivation, where raspberry bushes exhibited prominent viral symptoms (i.e., Vyhnánov), where almost all detections involved co-infection with two or more viruses. Analysis of co-infection patterns demonstrates that each commercial plantation and wild population harbored distinct virus combinations rather than a uniform profile. This heterogeneity can be explained not only by differences in virus transmission routes (i.e., pollen, aphids, and mites) but also by local management strategies, such as vector control, the quality of planting material, and the age of cultivated plants—particularly at individual cultivation sites and in the case of wild shrubs or plants from private gardens. Co-infection can lead to complex interactions with diverse outcomes that may be synergistic, antagonistic, or neutral. In synergistic cases—which are particularly concerning—virus transmission may become more efficient, and both the severity and impact of infection are exacerbated [59]. Despite ongoing efforts to understand within-host virus interactions, much remains poorly understood due to the unpredictable nature of the outcomes [59,60]. Furthermore, interactions between viruses and their respective vectors during mixed infection may further increase the variability and complexity of results [61]. Therefore, this highlights the need for a deeper understanding of mixed infections to accurately interpret the interactions and impact of these viruses on raspberry production in this study.

### 4.3. The Novel RaVA

The discovery of RaVA, together with two recently reported novel raspberry viruses—raspberry enamovirus 1 (RaEV1) [6] and raspberry rubodvirus 1 (RaRV1) [9]—highlights the importance of this study in monitoring viruses and arthropods on raspberry crops, despite the existence of extensive publications on the topic. RaVA is most related to recently described Cnidium virus Z (Acc. No. PP076733.1) from South Korea, but only shows moderate sequence identity—69.4% nucleotide and 74.4% aa identity in compared region of 484 nt [62]. Together with small differences among the isolates identified in this study, this indicates that the discovered virus represents an isolate of a new species. Our screening revealed that nearly 18% of the tested raspberry bushes were RaVA-positive. Notably, it was detected in all types of raspberry-growing sites, except for the two purchased plants. Its abundance in garden-collected plants is noteworthy, as the virus was found in more than half of the samples. It was also detected in wild raspberries and in *R. occidentalis*, all of which may act as reservoirs for infection. Further studies are essential to investigate this virus in greater detail, particularly its whole genome sequence, associated symptoms, transmission mechanisms, vectors, host plants, and its potential impact on raspberry growth.

### 4.4. Arthropods on Raspberry and Their Respective Virus

The finding that 20 out of the 27 identified phytophagous arthropod species had not previously been documented on raspberry [3] indicates that raspberry-associated arthropod diversity is likely underestimated, even though their pest status on this crop remains largely unknown. This pattern aligns with a recent thrips survey in Norway, where most detected species had also not been recorded on raspberry [63]. Both findings highlight the importance of surveying arthropods concurrently with viruses, as undocumented species can influence both perceived diversity and interpretations of virus–vector relationships. Examples from this study include *A. lineolatus* and *Frankiniella intonsa* (Trybom) (Thysanoptera: Thripidae), both previously undocumented on raspberry and detected with at least one of the tested viruses. *Adelphocoris lineolatus* was found carrying BRNV, RLMV, and RBDV in a single individual, but this does not reflect its vector status. Although this species has been associated with phytoplasma, its role as a vector remains unconfirmed [64,65]. The detection of multiple viruses may be related to its foraging behavior, as a single individual can feed on several raspberry plants during a foraging trip, potentially ingesting virus particles from multiple infected plants. It could also take up several viruses from a single plant, as the individual *A. lineolatus* analyzed here was collected at the Vyhnánov site, where most raspberry plants tested were co-infected with multiple viruses. While this foraging behavior has not been specifically studied in this species, it is common among mobile insects such as *A. lineolatus*. Moreover, as a generalist with a low host fidelity and a broad host range (at least 245 host plant species) [64,65,66], it may inadvertently introduce raspberry viruses to potential new hosts, given that this study had identified new natural and experimental hosts for RBDV and RLBV, respectively. In the case of *F. intonsa*, the species was not among the four previously undocumented thrips species recently reported in Norway [63], further supporting the underestimation of raspberry thrips diversity. It was also detected with BRNV. These findings suggest that transmission experiments are needed to determine the role of both species as potential vector of respective viruses.

Similarly, the detection of BRNV and RLMV by *A. idaei* may be related to its feeding behaviors, given that these viruses are vectored by another aphid, *A. rubi idaei*. A BRNV transmission experiment using *A. idaei* failed to infect the healthy recipients [41], suggesting that virus particles are ingested during phloem feeding without resulting in transmission. However, the relatively high number of BRNV-positive *A. idaei* samples indicates that further investigation is recommended, especially considering that efficient RVCV transmission, of which *A. idaei* is a known vector, occurs only in densely packed spring colonies and not in more dispersed summer colonies [50]. This observation implies that vector efficiency depends not only on species identity but also form of colony and seasonal timing, which may similarly influence BRNV transmission. Electron microscopy, previously used to confirm RVCV vector status of *A. idaei* [49], may help clarify the interaction between BRNV and *A. idaei*. RLMV detections in *A. idaei* largely occurred as co-infection with BRNV, consistent with earlier report [41], which may explained by co-ingestion of virus particles during phloem feeding. Similar approaches suggested for BRNV could clarify the interactions of RLMV and *A. idaei*.

A similar mechanism may account for BRNV detection in the leafhopper *Macropsis fuscula* (Zetterstedt) (Hemiptera: Cicadellidae). Although previously associated only with *Rubus* stunt phytoplasmas [67,68], *M. fuscula* is phloem feeder like aphids and may ingest BRNV particles while feeding. In addition to BRNV, *M. fuscula* was also found to carry RaEV1 in the recent study describing this newly identified raspberry virus [6]. Therefore, its potential role in the transmission of both BRNV and RaEV1 should be further investigated.

RBDV was detected in majority of the virus-positive arthropod species, including the known aphid vectors *A. rubi idaei* and *A. idaei*, as well as the predatory arthropods *Orius strigicollis* Poppius (Hemiptera: Anthocoridae) and *T.* (*T.*) *pyri*. This is likely because RBDV is pollen-borne, and its detection may simply result from pollen inadvertently landing on these arthropods. This is consistent with the ecology of these species. For instance, *A. idaei* colonies frequently occur near flower buds and new shoots, while both *Orius* spp. and *T.* (*T.*) *pyri* feed on pollen and nectar as alternative food sources. Increasing attention was given to the role of flower-visiting arthropods, such as pollinators, in the spread of pollen-borne plant viruses [46,47], although interactions vary across species. Understanding these interactions is important for raspberry production, which relies heavily on pollinators to maximize drupelet set, fruit quality, and overall marketability [69].

Generalist predators have been indispensable in supporting pesticide reduction efforts due to their flexible diets, which allow them to sustain populations on alternative food sources, such as pollens and nectars, even under low-prey conditions, thereby contributing to robust biological pest management strategies [70]. It is also because of this feeding flexibility, generalist predators, or omnivores—such as *Orius minutus* (Linnaeus) (Hemiptera: Anthocoridae) and *Psallus wagneri* Ossiannilsson (Hemiptera: Miridae) in this study—can also uptake plant viruses through feeding. The detection of RYNV in both species and RVCV in *O. minutus* may result from feeding on plant sap or on viruliferous arthropod prey, such as aphids [71,72,73]. Other than this study, RaEV1 was also detected in *P. wagneri* [6]. As pesticide reduction increases reliance on biological control agents, understanding how such species interact with plant viruses becomes increasingly important. In addition, *Psallus* spp. are typically strongly associated with their host plants [72], but no prior documentation has linked *P. wagneri* to *Rubus*. Therefore, further investigation of this relationship could be relevant for future biocontrol strategies, potentially providing an additional biocontrol option for raspberry pest management.

Finally, although spider mites have recently been shown to acquire RLBV [52], *N. rubi* was not found with any virus in this study, likely due to the low sample size.

## 5. Conclusions

This study provides the first comprehensive overview of raspberry viruses in the Czech Republic, encompassing all viruses listed in the EPPO Certification scheme, RLBV, and a novel virus—RaVA. The most frequently detected viruses were RBDV, BRNV, and RLMV. *Sambucus nigra* was identified as a new natural host of RBDV, and *N. occidentalis* 37B as a new experimental host of RLBV. Viruses occurred in plants and arthropods, both individually and in co-infection, and were also detected in 11 other arthropod species apart from the known vectors *A. rubi idaei* and *A. idaei*. The discovery of RaVA, the new host associations of RBDV and RLBV, and the detection of viruses in arthropod species—some previously undocumented on raspberry—broadens our understanding of raspberry virus diversity, host range, and potential vectors. In addition, 18 phytophagous arthropod species not found to carry viruses were recorded on raspberry for the first time; although their pest status remains uncertain, these findings expand our knowledge of the raspberry-associated arthropod community and provide a foundation for further studies. Overall, this study highlights the need for continued surveillance, improved diagnostics, and research on within-host virus interactions and virus–vector ecology to strengthen certification schemes and support sustainable raspberry production under changing environmental conditions.

## Figures and Tables

**Figure 1 viruses-17-01597-f001:**
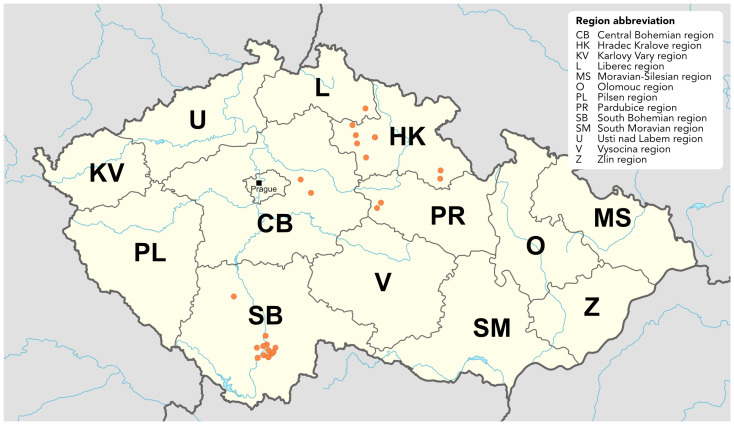
Overview of the sampling sites across the Czech Republic. Each sampling coordinate is indicated by an orange marker. Map adapted from work by NordNordWest, licensed under CC BY-SA 3.0 de. Source of original map: https://commons.wikimedia.org/w/index.php?curid=42830552, 1 May 2025.

**Figure 2 viruses-17-01597-f002:**
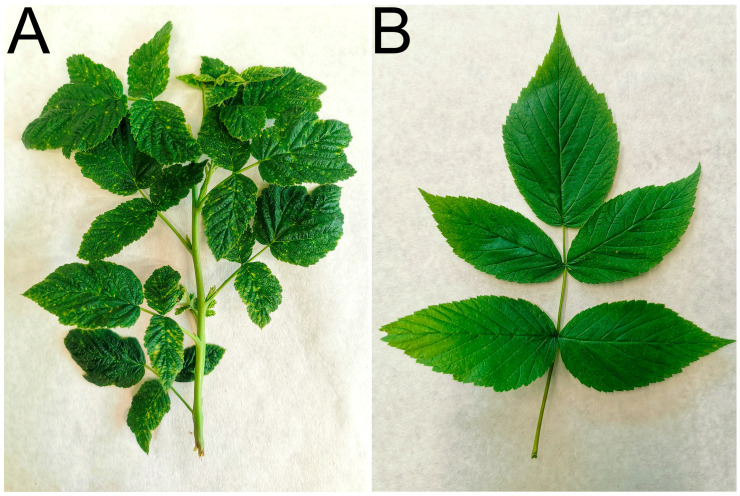
Representative examples of the sampled leaves and shoots: (**A**) a segment of a raspberry shoot exhibiting systemic, prominent chlorotic dots mosaic (sample B16, Dobré Pole; RLMV detected), and (**B**) a compound leaf consisting of five (occasionally three) leaflets showing very faint chlorotic dots and lines (sample B32, České Budějovice; BRNV, RBDV, and RaVA detected).

**Figure 3 viruses-17-01597-f003:**
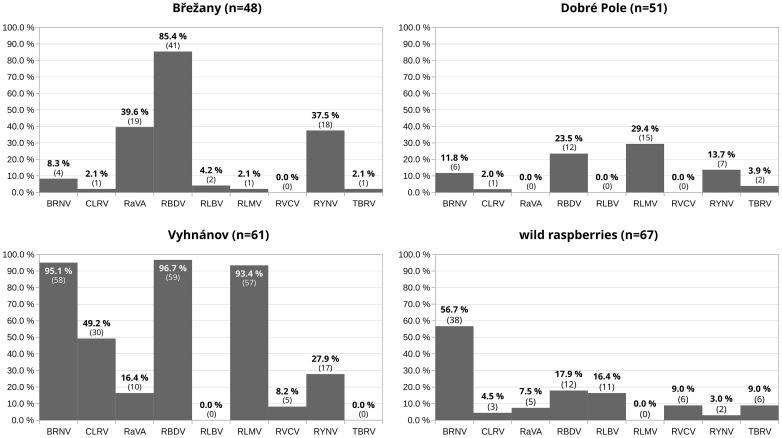
Prevalence of raspberry viruses at three selected commercial planting sites (with more than 40 samples tested) and in wild raspberry plants. The *y*-axis represents the percentage of plants infected at each location, and the *x*-axis indicates the nine detected viruses. For each bar, the percentage shows the exact prevalence of the corresponding virus, and the number of infected plants is given in parentheses.

**Figure 4 viruses-17-01597-f004:**
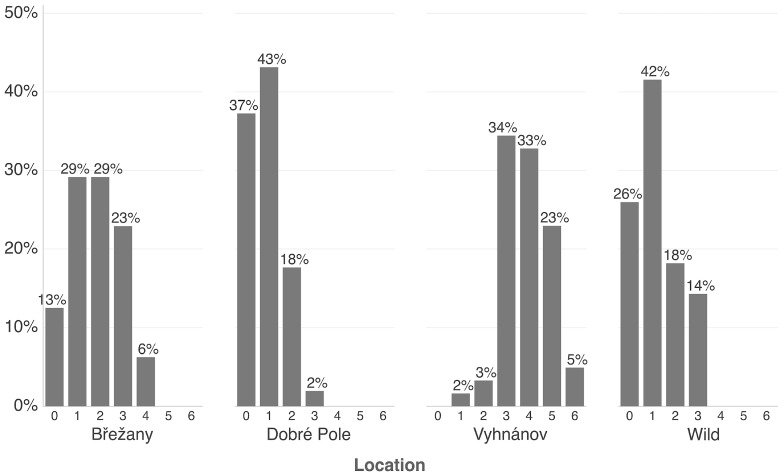
Percentage of raspberry plants (commercial and wild) grouped by the number of detected viruses, ranging from 0 (no virus detected) to 6 (co-infection with six viruses). The *y*-axis indicates the proportion (%) of plants in each category. Only commercial plantations with more than 40 tested samples are included.

**Figure 5 viruses-17-01597-f005:**
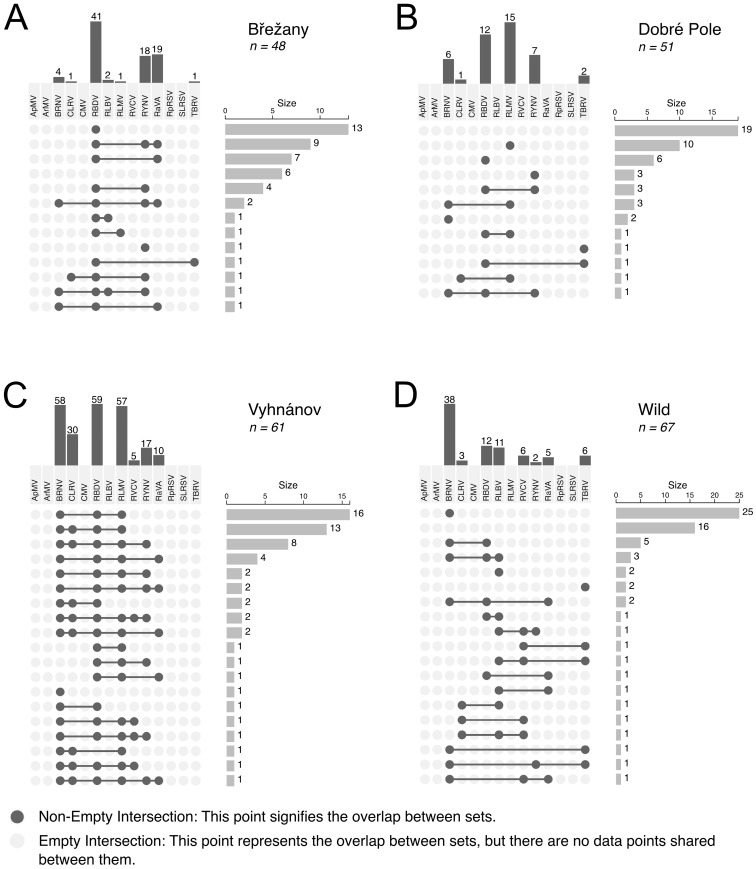
The upset plot showing the distribution and co-occurrence of virus infections in raspberry samples from selected commercial (**A**–**C**) and wild (**D**) sources. Only commercial plantations with more than 40 tested samples are included. Vertical bars indicate the number of samples in which the corresponding virus was detected, while horizontal bars represent the total number of samples in which either a virus or the corresponding co-infection of viruses was identified.

**Figure 6 viruses-17-01597-f006:**
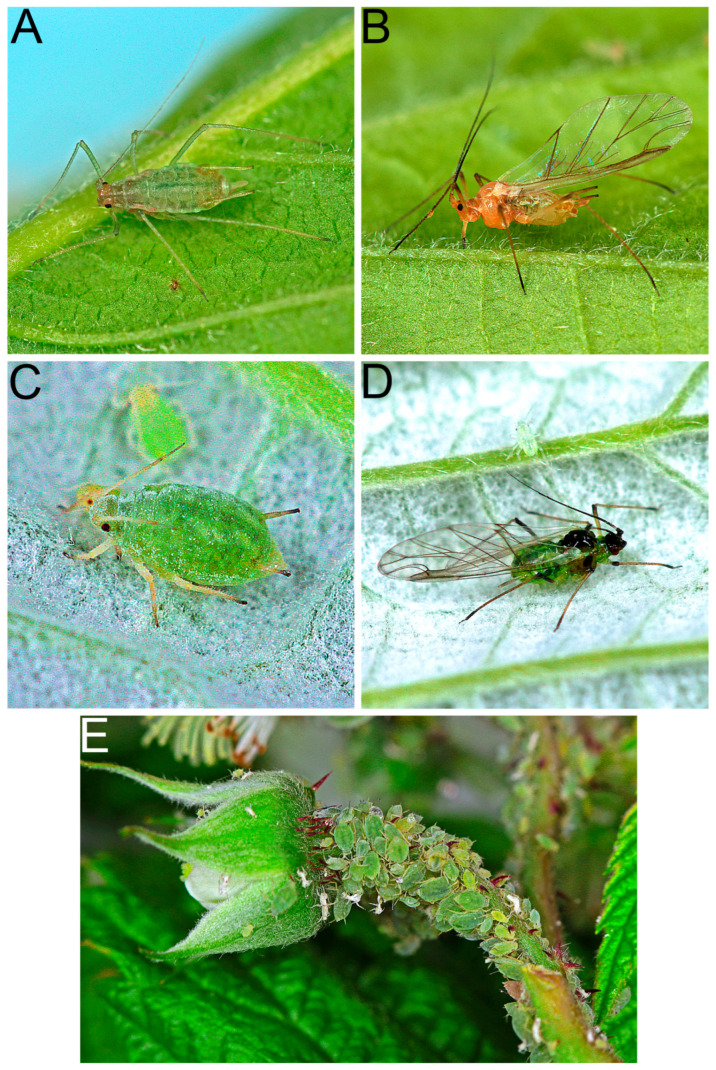
(**A**) Apterous of large raspberry aphid, *Amphorophora rubi idaei*; (**B**) Alate of large raspberry aphid, *A. rubi idaei*; (**C**) Apterous of small raspberry aphid, *Aphis idaei*; (**D**) Alate of small raspberry aphid, *A. idaei*; (**E**) A colony of small raspberry aphids, *A. idaei*, on the pedicel of a developing raspberry flower on a red raspberry plant, *R. idaeus*. Photos credited to Dr. J. Havelka.

**Figure 7 viruses-17-01597-f007:**
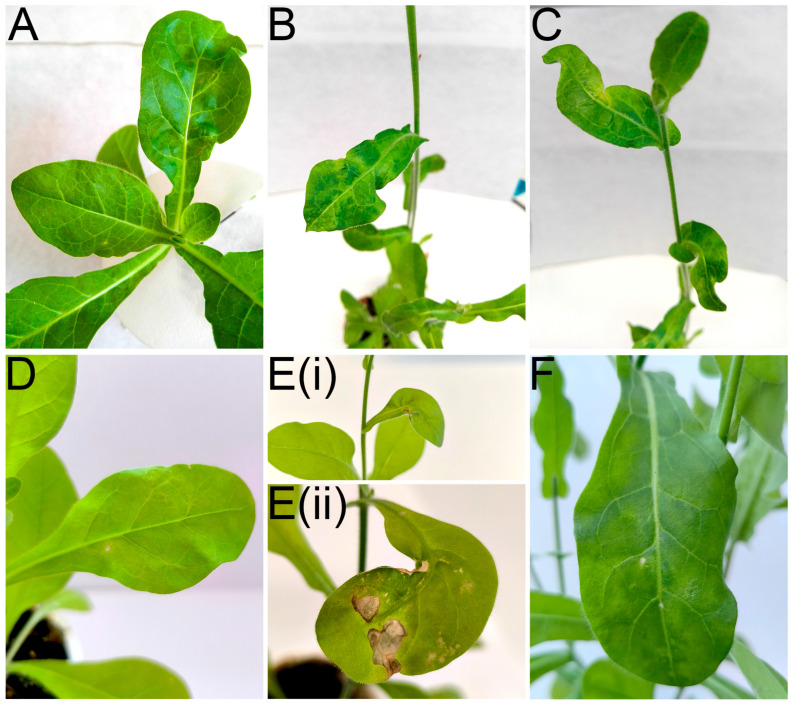
Symptom development on *Nicotiana occidentalis* 37B infected with RLBV. First transmission: (**A**) chlorosis and leaf deformation at 22 dpi; (**B**) leaf mosaic at 164 dpi; (**C**) leaf deformation at 164 dpi. Second transmission: (**D**) mild mosaic with necrotic spots at 24 dpi; (**E**) same leaf showing (i) severe deformation with necrosis at 30 dpi and (ii) worsening necrosis and deformation at 67 dpi; (**F**) dark green mosaic with a few chlorotic and necrotic spots at 113 dpi. Here, ‘dpi’ indicates days post-inoculation.

**Table 1 viruses-17-01597-t001:** The number of raspberry samples and sampling sites in five regions in the Czech Republic.

Type of Sample Site	Region	Number of Sites	Number of Samples
Commercial plantation	Central Bohemia	2	99
	Hradec Králové	3	71
Garden retailers	Liberec	1	1
	South Bohemia	1	1
Private garden	Hradec Králové	3	8
	Pardubice	2	2
	South Bohemia	5	8
Wild	Hradec Králové	1	6
	South Bohemia	6	61
Total		24	257

**Table 2 viruses-17-01597-t002:** Viruses tested in the plant samples.

Virus Name	Abbreviation	Family	Species
Apple mosaic virus	ApMV	*Bromoviridae*	*Ilarvirus ApMV*
Arabis mosaic virus	ArMV	*Secoviridae*	*Nepovirus arabis*
Cherry leaf roll virus	CLRV	*Secoviridae*	*Nepovirus avii*
Cucumber mosaic virus	CMV	*Bromoviridae*	*Cucumovirus CMV*
Black raspberry necrosis virus	BRNV	*Secoviridae*	*Sadwavirus rubi*
Raspberry-associated virus A	RaVA	n.a. ^1^	n.a. ^1^
Raspberry bushy dwarf virus	RBDV	*Mayoviridae*	*Idaeovirus rubi*
Raspberry leaf blotch virus	RLBV	*Fimoviridae*	*Emaravirus idaeobati*
Raspberry leaf mottle virus	RLMV	*Closteroviridae*	*Closterovirus macularubi*
Raspberry ringspot virus	RpRSV	*Secoviridae*	*Nepovirus rubi*
Raspberry vein chlorosis virus	RVCV	*Rhabdoviridae*	*Alphacytorhabdovirus alpharubi*
Rubus yellow net virus	RYNV	*Caulimoviridae*	*Badnavirus reterubi*
Strawberry latent ringspot virus	SLRSV	*Secoviridae*	*Stralarivirus fragariae*
Tomato black ring virus	TBRV	*Secoviridae*	*Nepovirus nigranuli*

^1^ species and family are not available yet, but it is suggested to be in the realm *Riboviria*, with genus *Ribovirus*.

**Table 3 viruses-17-01597-t003:** Number of samples tested for individual viruses, number of positive samples, and percentages by sample site origin.

Category of Raspberry Origin (Total Samples)	Virus Tested (Number of Positive; %)													
	ApMV	ArMV	BRNV	CLRV	CMV	RaVA	RBDV	RLBV	RLMV	RpRSV	RVCV	RYNV	SLRSV	TBRV
Commercial plantations (170)	0(0.0)	0(0.0)	68 (40.0)	32 (18.0)	0(0.0)	30 (17.6)	115 (67.6)	7 (4.1)	73 (42.9)	0(0.0)	6 (3.5)	42 (24.7)	0(0.0)	3 (1.8)
Garden retailer (2)	0(0.0)	0(0.0)	1 (50.0)	0(0.0)	1 (50.0)	0(0.0)	0(0.0)	0(0.0)	0(0.0)	0(0.0)	0(0.0)	1 (50.0)	0(0.0)	0(0.0)
Private garden (18)	0(0.0)	0(0.0)	1 (5.6)	2 (11.1)	1 (5.6)	11 (61.1)	6 (33.3)	2 (11.1)	0(0.0)	0(0.0)	8 (44.4)	4 (22.2)	0(0.0)	2 (11.1)
Wild (67)	0(0.0)	0(0.0)	38 (56.7)	3 (4.5)	0(0.0)	5 (7.5)	12 (17.9)	11 (16.4)	0(0.0)	0(0.0)	6 (9.0)	2 (3.0)	0(0.0)	6 (9.0)
Total (257)	0(0.0)	0(0.0)	108 (42.0)	37 (14.4)	2 (0.8)	46 (17.9)	133 (51.8)	20 (7.8)	73 (28.4)	0(0.0)	20 (7.8)	49 (19.1)	0(0.0)	11 (4.3)

For details see Appendix A.

**Table 4 viruses-17-01597-t004:** Arthropod samples positive for at least one of the six raspberry viruses.

Arthropod Species ^1^	TG ^2^	O ^3^	Total ^4^	Virus Detected ^5^					
				BRNV	RBDV	RLBV	RLMV	RVCV	RYNV
Hemiptera: Aphididae									
*Amphorophora rubi idaei* *	H	CP	31	17	4	0	8	0	0
		W	1	1	0	0	0	0	0
*Aphis idaei* *	H	CP	35	12	3	0	11	1	0
		G	7	0	1	0	0	2	0
		W	14	6	1	0	0	2	0
Hemiptera: Miridae									
*Adelphocoris lineolatus*	H	CP	1	1	1	0	1	0	0
Hemiptera: Cicadellidae									
*Macropsis fuscula* *	H	CP	3	1	0	0	0	0	0
Coleoptera: Curculionidae									
*Anthonomus rubi* *	H	CP	2	0	2	0	0	0	0
Thysanoptera: Thripidae									
*Frankliniella intonsa*	H	CP	1	1	0	0	0	0	0
*Thrips fuscipennis* *	H	CP	1	0	1	0	0	0	0
Hemiptera: Anthocoridae									
*Orius minutus*	P	CP	6	0	0	0	0	0	1
		G	1	0	0	0	0	1	0
		W	1	0	0	0	0	0	0
*Orius strigicollis*	P	G	1	0	0	0	0	0	0
		W	1	0	1	0	0	0	0
Hemiptera: Miridae									
*Psallus wagneri*	Om	W	1	0	0	0	0	0	1
Acari: Phytoseiidae									
*Typhlodromus* (*Typhlodromus*) *pyri*	P	CP	1	0	0	0	0	0	0
		G	1	0	1	0	0	0	0

^1,^* after the species name indicates species previously known to be present in raspberry [3]; ^2^ Trophic guild of the arthropod, H: Herbivore, P: Predator/Parasitoid, Om: Omnivore; ^3^ O is the category of origin of raspberry growing, where the arthropods were collected, CP: Commercial plantation; G: Garden; W: Wild-growing plants; ^4^ Total number of arthropod samples tested; ^5^ Number of arthropod samples detected with the respective viruses.

## Data Availability

All relevant data are included in this article, either within the main text or as Appendix A. The partial nucleotide sequence of raspberry viruses and *Phyllocoptes gracilis* are deposited in GenBank under the accession numbers: PX549241–PX549276 and PX444998, respectively. Additional data are available from the corresponding author upon reasonable request.

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
