# Peer review of "Raspberry Viruses in the Czech Republic, with Identification of a Novel Virus: Raspberry Virus A"

_viruses, 2025, doi:10.3390/v17121597_

Round 1

Reviewer 1 Report

Comments and Suggestions for Authors

This paper brings the results of the comprehensive research on the presence of viruses infecting raspberries and wild blackberries in the Czech Republic. In addition, the results of the virus presence in vectors (aphids and mites) are included in the study.

The study is of the great interest because it includes multidisciplinary research, including plant virology and entomology. The results of the survey for novel virus (RaVA) brought new knowledge on the existence of this newly discovered virus on raspberries. Transmission study of RLBV to herbaceous plant is a novelty in the research of this virus. Also, the finding of the 20 phytophagous arthropods that nave not previously been recorded on raspberries is significant and a good base for further research on their pest and vector status.

All parts of the manuscript are elaborated in detail.

Introduction part is proper for the topic.

Material and methods section is written in detail, allowing other authors to access into the details of each performed step.

Results are nicely presented. Text is adequate and tables and figures are representative.

Discussion is well elaborated, comparing the obtained result with adequate references.

I do not have any major remarks on the manuscript.

Some minor issues:

Line 332. Accession number is missing

Lines 440-442. Proper citations for this statement are:

[38] https://doi.org/10.1111/aab.12247

https://doi.org/10.5424/sjar/2019171-13861

Line 464: ApMV was confirmed in red and black raspberry.

Correct this.

Line 480: ongoing

Author Response

For research article

Response to Reviewer 1 Comments

1. Summary

The authors would like to thank the reviewer for his/her careful and thoughtful evaluation of the manuscript. We are grateful for his/her valuable comments emphasizing the importance of this work and for his/her constructive suggestions to improve its clarity and accuracy. All suggestions have been accepted, and the necessary revisions have been made accordingly. Please find the detailed response below and the corresponding revisions in track changes in the re-submitted files.

2. Questions for General Evaluation

Reviewer’s Evaluation

Response and Revisions

Does the introduction provide sufficient background and include all relevant references?

Yes

We would like to thank you the reviewer again.

Are all the cited references relevant to the research?

-

Is the research design appropriate?

Yes

Are the methods adequately described?

Yes

Are the results clearly presented?

Yes

Are the conclusions supported by the results?

Yes

3. Point-by-point response to Comments and Suggestions for Authors

Comments 1: Line 332. Accession number is missing

Response 1: The accession number has been added to the manuscript

Comments 2: Lines 440-442. Proper citations for this statement are:

[38] https://doi.org/10.1111/aab.12247

https://doi.org/10.5424/sjar/2019171-13861

Response 2: Agree, we have revised it as suggested. However, due to the revision, the line number may now have changed.

Comments 3: Line 464: ApMV was confirmed in red and black raspberry.

Correct this.

Response 3: Agree, we have revised it as suggested. We are grateful to the reviewer for pointing out an important fact that we have overlooked.

Comments 4: Line 480: ongoing

Response 4: Agree, we have revised it as suggested.

4. Response to Comments on the Quality of English Language

The English is fine and does not require any improvement.

Response: We are grateful for the reviewer recognizing our language competency. Therefore, we did not proceed for any English editing.

5. Additional clarifications

N/A

Reviewer 2 Report

Comments and Suggestions for Authors

I made all the comments and suggestion in the attached pdf version of the manuscript.

Author Response

For research article

Response to Reviewer 2 Comments

1. Summary

The authors would like to thank the reviewer for his/her careful and thoughtful evaluation of the manuscript. We are grateful for the valuable comments and constructive suggestions to improve its clarity and accuracy. We have made every effort to revise the manuscript wherever possible at this stage. Please find the detailed response below and the corresponding revisions in track changes in the re-submitted files.

2. Questions for General Evaluation

Reviewer’s Evaluation

Response and Revisions

Does the introduction provide sufficient background and include all relevant references?

Can be improved

We have improved it.

Are all the cited references relevant to the research?

-

This was not commented

Is the research design appropriate?

Must be improved

We have included the requested detail

Are the methods adequately described?

Must be improved

We have included as much detail as possible

Are the results clearly presented?

Can be improved

We have improved wherever commented

Are the conclusions supported by the results?

Can be improved

Only requested to be concise, which we have revise accordingly

3. Point-by-point response to Comments and Suggestions for Authors

Comments 1: either write this in Czech context or global.

Response 1: We added the word global to clarify that it is about global production of raspberry.

Comments 2: Introduction section lacks a clear articulation of research gaps or hypotheses. It mostly repeats known background information without stating what this study adds beyond previous Czech surveys

Response 2: The introduction has been revised with expanded literature on raspberry virus research in the Czech Republic and supplemented with information on new developments compared to previous surveys. Research gaps and hypotheses have also been highlighted to underscore the significance of this study.

Comments 3: should be “Raspberry (Rubus idaeus L.) is known to be affected

Response 3: Revised accordingly

Comments 4: italicize.

Response 4: According to The International Commission on Zoological Nomenclature: The names of higher-ranking groups e.g. families or orders always begin with a capital but are not italicised. https://www.iczn.org/assets/dfa53e9e0c/ICZN-Advice-for-editors-of-popular-works-web-v10-_.pdf

Although you may be confused with those for viruses, which is different: https://ictv.global/faq/names

Comments 5: Better show the figures of the collected leaves.

Response 5: Figure added as Figure 2 under section 2.1

Comments 6: In which months?

Response 6: Revised

Comments 7: The presentation of the script would be significantly improved by showing this in a figure.

Response 7: A map with plotted sampling site locations has been added as Figure 1.

Comments 8: Were these plants without the insect infestation?

Response 8: It is without insect infestation, and we have revised the sentence to make it absolute clear.

“…asymptomatic and arthropod-free plants ….”

Comments 9: why 257 samples were deemed sufficient? No methodology for sample and field selection is mentioned.

Response 9: A total of 257 plant samples were collected because of a targeted and pragmatic sampling strategy designed to maximize the likelihood of detecting viral infections (for e.g., purposively selecting plant with suspected viral symptoms, and those infested by known or potential vector) while considering logistical constraints. Sampling focused primarily on areas of the major raspberry growers in the Czech Republic (Dobré Pole, BÅ™ežany), supplemented by smaller growers who either requested examination due to observed disease symptoms (Kbelnice, Synkov) or permitted sampling (Vyhnánov). Preference was always given to plants exhibiting symptoms consistent with viral diseases and/or the presence of arthropod vectors. To capture potential latent infections, asymptomatic plants were also collected.

Comments 10: This subsection does not provide enough detail for the collection sites and strategy arthropods collection.

Response 10: Revised the subsection to include more details on collection and sampling method.

Comments 11: Authors can mention accession number here, if available.

Response 11: Although we have sequences from HTS for this isolate of novel RaVA, the entire sequence is not yet available and therefore we do not include it here.

Comments 12: Details on RT-qPCR primer validation, sensitivity, and controls are insufficient for reproducibility.

Response 12: Thank you for your comment. We have clarified the details in Section 2.3.3, including primer validation and the controls used.

Primer validation: All primers were tested for specificity using melt-curve analysis, ensuring single peaks and the absence of primer dimers. The amplified products were further confirmed by Sanger sequencing.

Controls: No-template controls and positive controls were included in every run to monitor contamination and verify amplification. An endogenous control for NADH and mRNA was used to validate cDNA synthesis and subsequent qPCR amplification.

A quantitative estimation of assay sensitivity would require a different experimental approach, such as the use of synthetic templates, which was beyond the scope of the current study.

Comments 13: Better mention the quantity.

Response 13: We have added the quantity of the original RNA. “Prepared as previously described” refers to the cDNA synthesis from RNA, which is detailed in subsection 2.3.1—it is now explicitly stated in the sentence. The tenfold-diluted cDNA refers to a tenfold dilution of this previously synthesized cDNA, from which 5 μL was used in each reaction. We hope this clarifies the procedure. The exact quantity of tenfold-diluted cDNA cannot be explicitly stated, as it depends on the RNA input and other factors affecting cDNA synthesis.

Comments 14: Transmission experiments lack replicates and control treatments

Response 14: Although transmission experiments lack replicates, in both cases Phyllocoptes gracilis was observed on RLBV-positive raspberry plants exhibiting severe leaf blotch symptoms—same plants used for transmission—and the virus was successfully transmitted to N. occidentalis 37B. At later stages of infection, both infected N. occidentalis plants developed similar mosaic symptoms.

Unfortunately, we did not have access to virus-free Phyllocoptes gracilis for use in control transmissions. In 2021, we used Aphis idaei (aphid species) from our virus-free breeding stock as a control, where P. gracilis was absent in these controls. All then N. occidentalis 37B plants (n=10) remained symptom-free. This suggests that the virus is only transmitted in the present of P. gracilis.

Therefore, based on these observations, we believe this represent an important finding, despite the experiments not being typical transmission assays due to the difficulty of handling P. gracilis as well as RLBV.

Comments 15: Were the same leaves used for the RT-PCR assay? How was the presence of the virus confirmed in these symptomatic leaves?

Response 15: Unfortunately, the leaves used for RT-PCR to confirm the presence of RLBV were not the same as those used to transmit RLBV to N. occidentalis 37B. However, in the first attempt in 2021, tissues from leaves growing on the same shoot, approximately 2–3 cm below the leaves use for transmission, were used for RT-PCR.

The plant used for RLBV transmission in the second attempt (2025) exhibited severe leaf blotch symptoms for several consecutive years and consistently tested positive for RLBV. It was moved from the forest to our institution’s experimental field in 2022 and has been continuously monitored for RLBV since then. Symptomatic leaves were collected from this plant in 2022, 2023, 2024, and 2025, and they consistently tested positive for RLBV using RT-PCR. Therefore, we are confident that the symptomatic leaves collected for transmission in 2025 were also positive for RLBV, given the consistent RT-PCR results.

Comments 16: Better correlate symptoms and presence of multiple viruses in a plant/location.

Response 16: The current methodology does not allow us to draw any rigorous correlations. Without knowledge of co-infecting agents (with HTS), the symptoms are unreliably attributed, making any resulting conclusion too speculative.

Nevertheless, we have added observations of symptoms at individual locations in relation to virus presence in the Results section. To illustrate the diversity of symptoms, we added Figure S1 to the Supplementary Materials, showing various symptoms observed in 19 different plants. In addition, two representative examples of symptomatic samples have been added as Figure 2. We also compared the observed symptoms with descriptions reported in the literature in the Discussion section.

Comments 17: Add more details to the figure legends to enhance clarity.

Response 17: Unfortunately for the pack graph, we are unable to add in more legends while keeping a sufficiently large graph for readers. However, we have understood your concern, and have added more description in the caption to make up for the lesser information in the graph. We hope for your understanding.

Comments 18: Explain the numbers (values) mentioned in each bar for the clarity of the readers.

Response 18: We have added an explanation of these numbers in the figure caption, as follows:

“Vertical bars indicate the number of samples in which the corresponding virus was detected, while horizontal bars represent the total number of samples in which either a virus or the corresponding a co-infection of viruses was identified.”

Comments 19: Punctuation

Response 19: Corrected.

Comments 20: Italicize here, and wherever is applicable

Response 20: Same as comment 4, according to The International Commission on Zoological Nomenclature: The names of higher-ranking groups e.g. families or orders always begin with a capital but are not italicised. https://www.iczn.org/assets/dfa53e9e0c/ICZN-Advice-for-editors-of-popular-works-web-v10-_.pdf

Although you may be confused with those for viruses, which is different: https://ictv.global/faq/names

Comments 21: Figures are quite large in size.

Response 21: We have reduced the size while keeping it sufficiently clear for interested readers.

Comments 22: Mention Acc numbers.

Response 22: It has been added.

Comments 23: How many number of plants? Were all the plant showing symptoms?

Response 23: There was only one plant during the first attempt, and in the second attempt there are 15 Nicotiana occidentalis and 15 Physalis floridana. Among the 15 N. occidentalis, only one was observed with symptoms.

Comments 24a: The Upset plot and prevalence graphs are informative but inadequately referenced in the discussion.

Response 24a: We have revised and expanded the Discussion section to address this point comprehensively. A new paragraph was added that explicitly interprets the co-infection patterns revealed by the upset plot (Figure 3), specifically detailing the heterogeneity of virus combinations found across different commercial and wild Rubus populations.

Comments 24b: Discussion mostly restates results, while deeper mechanistic insights and comparison with global trends are missing.

Response 24b: We have improved and revised the discussion to avoid only restating the results.

Comments 25: Use abbreviation

Response 25: Revised

Comments 26: Concise this section

Response 26: We have made our best effort to streamline the conclusion while preserving the key message.

4. Response to Comments on the Quality of English Language

The reviewer selected “The English is fine and does not require any improvement”

Response: Therefore, language is maintained, and did not proceed for any English editing services.

5. Additional clarifications

N/A

Round 2

Reviewer 2 Report

Comments and Suggestions for Authors

Authors addressed all the raised ueries/comments/suggestions.